# SKDCGN: Source-free Knowledge Distillation of Counterfactual Generative Networks using cGANs

**Abstract.** With the usage of appropriate inductive biases, Counterfactual Generative Networks (CGNs) can generate novel images from random combinations of shape, texture, and background manifolds. These images can be utilized to train an invariant classifier, avoiding the wide spread problem of deep architectures learning spurious correlations rather than meaningful ones. As a consequence, out-of-domain robustness is improved. However, the CGN architecture comprises multiple over parameterized networks, namely BigGAN and U2-Net. Training these networks requires appropriate background knowledge and extensive computation. Since one does not always have access to the precise training details, nor do they always possess the necessary knowledge of counterfactuals, our work addresses the following question: Can we use the knowledge embedded in pre-trained CGNs to train a lower-capacity model, assuming black-box access (i.e., only access to the pretrained CGN model) to the components of the architecture? In this direction, we propose a novel work named SKDCGN that attempts knowledge transfer using Knowledge Distillation (KD). In our proposed architecture, each independent mechanism (shape, texture, background) is represented by a student 'TinyGAN' that learns from the pretrained teacher 'BigGAN'. We demonstrate the efficacy of the proposed method using state-of-the-art datasets such as ImageNet, and MNIST by using KD and appropriate loss functions. Moreover, as an additional contribution, our paper conducts a thorough study on the composition mechanism of the CGNs, to gain a better understanding of how each mechanism influences the classification accuracy of an invariant classifier. Code available at: https://anonymous.4open.science/r/SKDCGN-E753/README.md

## 1 Introduction

Deep neural networks are prone to learning simple functions that fail to capture intricacies of data in higher-dimensional manifolds [1], which causes networks to struggle in generalizing to unseen data. In addition to spectral bias [1] and shortcut learning, which are properties inherent to neural networks [2], spurious learned correlations are also caused by biased datasets. To this end, Counterfactual Generative Networks (CGNs), proposed by [3], have been shown to generate novel images that mitigate this effect. The authors expose the causal structure of image generation and split it into three Independent Mechanisms (IMs) (object shape, texture, and background), to generate synthetic and *counterfactual* images whereon an invariant classifier ensemble can be trained.

The CGN architecture comprises multiple over-parameterized networks, namely BigGANs [4] and U2-Nets [5], and its training procedure generally requires appropriate domain-specific expertise. Moreover, one does not always have access to the precise training details, nor do they necessarily possess the required knowledge of counterfactuals. Motivated by these observations, we propose *Source-free Knowledge Distillation of Counterfactual Generative Networks* (SKDCGN), which aims to use the knowledge embedded in a pre-trained CGN to train a lower capacity model, assuming black-box access (i.e., only inputs and outputs) to the components of the source model. More specifically, we harness the idea of Knowledge Distillation (KD) [6] to train a network comprising three (small) generative models, i.e. TinyGANs [7], each being responsible for a single independent mechanism. SKDCGN carries both practical and theoretical implications, and it is intended to:

1. Obtain a lightweight version of the CGN, reducing its computational cost and memory footprint. This is meant to (i) ease the generation of counterfactual datasets and hence encourage the development of robust and invariant classifiers, as well as (ii) potentially allowing the deployment of the model on resource-constrained devices.
2. Explore whether we can *learn* from a fully trained CGN and distill it to a less parameterized network, assuming that we do not have access to the training process of the model.

Along the lines of the original paper, we demonstrate the ability of our model to generate counterfactual images on ImageNet-1k [8] and Double-Colored MNIST [3]. Furthermore, we compare our outputs to [3] and a simple baseline in terms of out-of-distribution robustness on the original classification task. As an additional contribution, we conduct a study on the shape IM of the CGN.

The paper is organized as follows: firstly, we present a brief literature survey in Section 2; next in Section 3 the SKDCGN is dissected; Section 4 presents the experimental setup and the empirical results, which are finally discussed in Section 5.

## 2   Related work

This section introduces the fundamental concepts and the related works that we use as a base for our SKDCGN.

**Counterfactual Generative Networks** The main idea of CGNs [3] has already been introduced in Section 1. Nonetheless, to aid the understanding of our method to readers that are not familiar with the CGN architecture, we summarize its salient components in this paragraph and also provide the network diagram in Appendix Section A.1 Figure 1. The CGN consists of 4 backbones: (i) the part of the network responsible for the shape mechanism, those responsible for (ii) texture and (iii) background, and a (iv) composition mechanism that combines the previous three using a deterministic function. Given a noise vector $\mathbf{u}$ (sampled from a spherical Gaussian) and a label $y$ (drawn uniformly from the set

of possible labels y) as input, (i) the shape is obtained from a BigGAN-deep-256 [4], whose output is subsequently passed through a U2-Net [5] to obtain a binary mask of the object shape. The (ii) texture and (iii) background are obtained similarly, but the BigGAN's output does not require to be segmented by the U2-Net. Finally, the (iv) composition mechanism outputs the final counterfactual image $\mathbf{x}_{gen}$ using the following analytical function:

$$\mathbf{x}_{gen} = C(\mathbf{m}, \mathbf{f}, \mathbf{b}) = \mathbf{m} \odot \mathbf{f} + (1 - \mathbf{m}) \odot \mathbf{b}, \tag{1}$$

where $\mathbf{m}$ is the shape mask, $\mathbf{f}$ is the foreground (or texture), $\mathbf{b}$ is the background and $\odot$ denotes element-wise multiplication.

More recently, [9] devises an approach that learns a latent transformation that generates visual CFs automatically by steering in the latent space of generative models. Additionally, [10] uses a deep model inversion approach that provides counterfactual explanations by examining the area of an image.

**Knowledge Distillation**. [11] firstly proposed to transfer the knowledge of a pre-trained cumbersome network (referred to as the *teacher*) to a smaller model (the *student*). This is possible because networks frequently learn low-frequency functions among other things, indicating that the learning capacity of the big network is not being utilized fully [1] [2]. Traditional KD approaches (often referred to as *black-box*) simply use the outputs of the large deep model as the teacher knowledge, but other variants have made use of activation, neurons or features of intermediate layers as the knowledge to guide the learning process [12,13]. Existing methods like [7] are also making use of Knowledge distillation for the task of image generation. Our work is similar to this, however, they transfer the knowledge of BigGAN trained on ImageNet dataset to a TinyGAN. In contrast, in our work, we transfer not just the knowledge of image generation but also the task of counterfactual generation from a BigGAN to a TinyGAN.

*Distilling GANs using KD.* Given its high effectiveness for model compression, KD has been widely used in different fields, including visual recognition and classification, speech recognition, natural language processing (NLP), and rec-ommendation systems [14]. However, it is less studied for image generation. [15] firstly applied KD to GANs. However, our project differs from theirs as they use *unconditional* image generation, less general (DCGAN [16]) architectures and they do not assume a black-box generator. Our setting is much more similar to that of [7], where a BigGAN is distilled to a network with $16\times$ fewer parameters, assuming no access to the teacher's training procedure or parameters. Considering its competitive performance, we use the proposed architecture (TinyGAN) as the student model and use a modified version of their loss function (further details in Section 3.1) to optimize our network.

**Source-free**: We term our method as Source-free since we do not have access to the source data, source training details, procedure, and any knowledge about the counterfactuals, etc, but only have access to trained source models. This method is similar to methods such as [17] [18]. With large diffusion models like Imagen [19] and DALL·E 2 [20] where the training process is usually extremely

expensive in terms of computation, lack precise details about training them and often not reproducible by academic groups, we often have access to pretrained models. These can be used to transfer knowledge to a smaller network, and perform the same task with model of lower capacity.

## 3   Approach

This section dives into the details of the SKDCGN architecture, focusing on the training and inference phases separately for ImageNet-1k and MNIST. In addition, we discuss the loss functions that were employed for Knowledge Distillation.

### 3.1   SKDCGN

Transferring the knowledge of an entire CGN into a single generative model could drastically reduce the number of parameters, however this strategy compromises the whole purpose of CGNs, i.e. disentangling the three mechanisms and having control over each of them. Therefore, we opt to train a generative model for each individual component. As shown in the architecture diagram (Figure 1), we treat each IM backbone as a black-box teacher and aim to mimic its output by training a corresponding TinyGAN student. Note that this implies that in the case of the shape mechanism, a single generative model learns to mimic both the BigGAN and the U2-Net. We believe a TinyGAN should be capable of learning binary masks directly, removing the need for the U2-Net and reducing the model size even further. During inference, the outputs of the three students are combined into a final counterfactual image using the composition function of Equation 1.

**Training: Distilling the knowledge of IMs**   To train SKDCGN, we utilize each IM backbone from the CGN architecture as a black-box teacher for the student network, as visualized in the training section of Figure 1 (the backbones are BigGAN + U2-Net for *shape*, BigGAN for *texture*, and BigGAN for *background*). Refer to Appendix Section A.2 Figure 2 for details about the training data generation. As introduced in the Related work section, [7] proposed an effective KD framework for compressing BigGANs. As the IMs in CGNs rely on BigGANs, we utilize their proposed student architecture. For completeness, the student architecture details are reported in Appendix Section A.2 Figure 2a.

We base our training objective on the loss function proposed by [7]. Our full objective comprises multiple terms: (i) a pixel-wise distillation loss, (ii) an adversarial distillation loss, (iii) a feature-level distillation loss, and (iv) KL Divergence. In addition to introducing KL Divergence, we deviate from the original TinyGAN training objective by omitting the term that allows the model to learn from real images of the ImageNet dataset. This would inevitably compromise the quality of the generated counterfactuals. KL Divergence leads to entropy minimization between the teacher and student, which is why we propose its usage.

The individual loss terms are dissected below as from [7]:

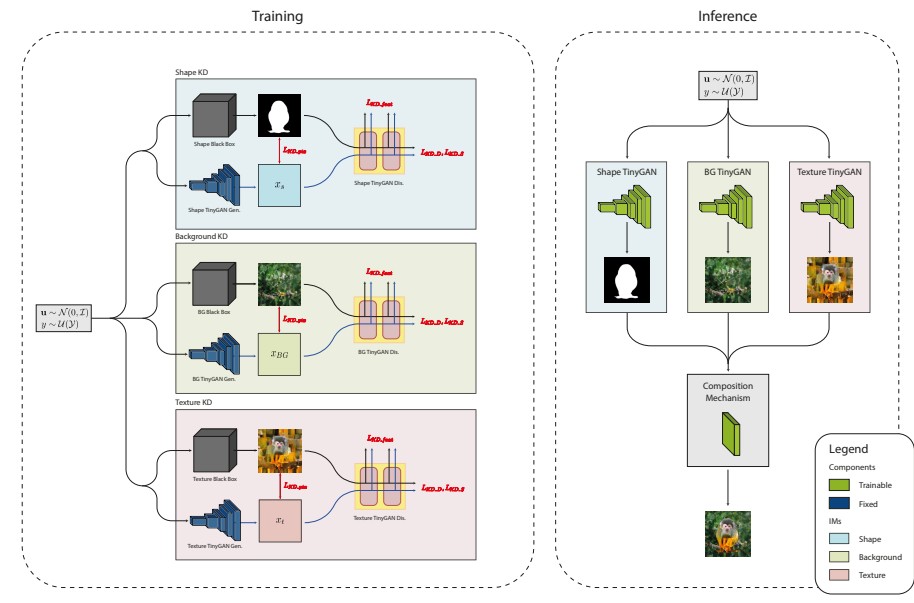

Fig. 1: *Architecture of the SKDCGN.* During training, each independent mechanism serves as a black-box teacher model to train a corresponding student model. During inference, the outputs of the three trained TinyGANs are combined using a Composition Mechanism that returns the final counterfactual image.

1. *Pixel-wise Distillation Loss*: To imitate the functionality of BigGAN for scaling generation to high-resolution, high-fidelity images, we minimize the pixel-level distance (L1) between the images generated by BigGAN and TinyGAN given the same input:

$$\mathcal{L}_{\text{KD\_pix}} = \mathbb{E}_{z \sim p(z), y \sim q(y)}[\|T(z,y) - S(z,y)\|_1] \qquad (2)$$

where $T$ represents the Teacher network, $S$ represents the Student network, $z$ is a latent variable drawn from the truncated normal distribution $p(z)$, and $y$ is the class label sampled from some categorical distribution $q(y)$.

2. *Adversarial Distillation Loss*: To promote sharper outputs, an adversarial loss is incorporated to make the outputs of $S$ indistinguishable from those of $T$. It includes a loss for the generator (Eq. 3) and one for the discriminator (Eq. 4):

$$\mathcal{L}_{\text{KD\_S}} = -\mathbb{E}_{z,y}[D(S(z,y),y)] \qquad (3)$$
$$\mathcal{L}_{\text{KD\_D}} = -\mathbb{E}_{z,y}[max(0, 1 - D(T(z,y),y)) + max(0, 1 - D(S(z,y),y))], \qquad (4)$$

where $z$ is the noise vector, $y$ is the class label, $T(z,y)$ is the image generated by the Teacher $T$, while $S$ and $D$ respectively are the generator and discriminator of the Student $S$.

3. *Feature Level Distillation Loss*: To further overcome the blurriness in the images produced by the Student network, the training objective also includes a feature-level distillation loss. More specifically, we take the features computed at each convolutional layer in the Teacher discriminator, and with a loss function stimulate $S$ to generate images similar to $T$:

$$\mathcal{L}_{\text{KD\_feat}} = \mathbb{E}_{z,y}\left[\sum_i \alpha_i \left\|D_i(T(z,y),y) - D_i(S(z,y),y)\right\|_1\right] \quad (5)$$

where $D_i$ represents the feature vector extracted from the $i^{th}$ layer of the discriminator and the corresponding weights are given by $\alpha_i$.

4. *KL Divergence*: L1 alone cannot reduce the entropy between the teacher and target. To improve the existing method, we use KL Divergence in a similar fashion to [21] for the task of Knowledge Distillation between real images drawn from source $P(x)$ and target images $Q(x)$.

$$\mathcal{D}_{\text{KL}}(P\|Q) = \sum_{x\in\mathcal{X}} P(x)\log\left(\frac{P(x)}{Q(x)}\right) \quad (6)$$

To sum up, the student generator $(S)$ and discriminator $(D)$ are respectively optimized using the following objectives:

$$\mathcal{L}_S = \mathcal{L}_{\text{KD\_feat}} + \lambda_1\mathcal{L}_{\text{KD\_pix}} + \lambda_2\mathcal{L}_{\text{KD\_S}} \quad (7)$$
$$\mathcal{L}_D = \mathcal{L}_{\text{KD\_D}} \quad (8)$$

where $\lambda_1$ and $\lambda_2$ are the regularization terms mentioned in [7].

Finally, note that the original CGN architecture (Appendix Section A.1 Figure 1) comprises another BigGAN trained on ImageNet-1k, which is unrelated to the three Independent Mechanisms and provides primary training supervision via reconstruction loss. We discard this component of the architecture for two main reasons: we do not have a dataset of counterfactuals whereon a GAN can be trained; we argue that this additional knowledge is already embedded in the backbones of a pre-trained CGN.

**Inference: generating counterfactuals** Once the three student networks are trained, their outputs are combined during inference akin to [3] using the analytical function 1. Since the composition function is deterministic, we devise inference as a separate task to training.

## 4    Experiments and results

In this section, we first define our experimental setup, then present our results. We test SKDCGN on ImageNet-1k (Section 4.3), and based on the observed findings we make some changes to the proposed architecture to improve the quality of the results (Section 4.4). Due to the computational constraints, we test these improvements on MNIST [22]. Finally, we present the results of our study of the shape IM (Section 4.5).

### 4.1   Datasets

*ImageNet-1k.* The ImageNet-1k ILSVRC dataset [8] contains 1,000 classes, with each class consisting of 1.2 million training images, 50,000 validation and 100,000 test images. Images were resized to $256 \times 256$ to maintain consistent experiments and to allow direct comparisons with the original results of [3].

*Double-colored MNIST.* We use the *double-colored* MNIST dataset proposed by Sauer and Geiger in the original CGN paper [3]. This is a variant of the MNIST dataset where both the digits and the background are independently colored. It consists of 60,000 $28 \times 28$ images of the 10 digits, along with a test set of 10,000 images.

### 4.2   Baseline Model: CGN with generator replaced by TinyGAN generator

The SKDCGN is compared with a modified version of the original CGN architecture, where each BigGAN has been replaced by the generator model of a TinyGAN. Training this baseline using the procedure described by [7], omitting KD, allows for rigorous comparisons that emphasize the effectiveness of the knowledge distillation process. Further training details are provided in Appendix Section C.1.

### 4.3   Results of SKDCGN

During training, we found that the TinyGANs could closely approximate the output in each IM. However, we observed that the trained TinyGANs could not generalize when given random noise to the generator to produce results beyond the test set. This may be due to the reduced network capacity of the TinyGAN model. Furthermore, each TinyGAN was trained on all 1000 classes in ImageNet-1K, as opposed to just 397 classes [7] used. We generate the test samples using the test and do not make use of random noise to generate the data. Since we believe that the student (our architecture) has only learned the specifics manifolds as that of the teacher (CGN). Therefore, when we use random noise we are not entirely sure if that is being generated from the same manifold as the teacher.

In Figure 2 we compare the outputs of the CGN backbone responsible for different independent mechanisms and those of the corresponding TinyGAN, given the same input.

We also make use of the Double-colored MNIST dataset to validate our method similar to ImageNet. Similarly, we make use of 3 IMs to train using our architecture as described in Figure 1. We obtain results as shown Appendix Section B.3 in Figures 4a, 4b, and 4c. We observe that our architecture is able to generate the corresponding IMs.

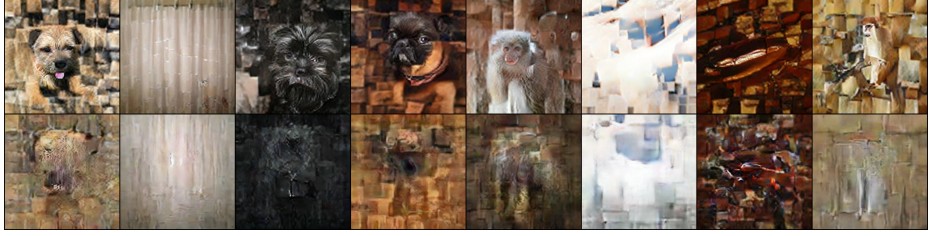

(a) A comparison of images generated by the CGN **shape** backbone (*top* row) and those generated by the corresponding SKDCGN given the same input (*bottom* row).

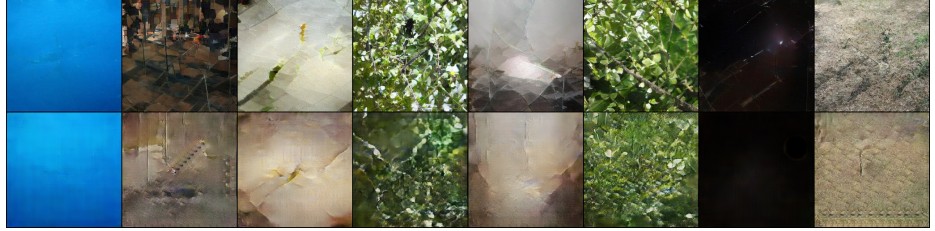

(b) A comparison of images generated by the CGN **texture** backbone (*top* row) and those generated by the corresponding SKDCGN given the same input (*bottom* row).

(c) A comparison of images generated by the **background** backbone (*top* row) and those generated by the corresponding SKDCGN given the same input (*bottom* row).

Fig. 2: A comparison of images generated by the CGN backbones and those generated by the corresponding SKDCGN (given the same input) for each independent mechanism.

### 4.4   Improving the SKDCGN

We realize that the student is as good as the teacher. We observe that the outputs are noisy and ambiguous in nature when generated using weights given by the authors of CGN [3]. Therefore, we observe several artifacts in the outputs generated by our architecture as well. Interestingly, while investigating the influence of Shape IM for MNIST, we observed that when the mask component was made smoother/transparent by using 3/4th of the mask, we observed an increase in the accuracy of CGN's invariant classifier than what was reported in the CGN paper. This suggests that we mask component of MNIST is noisy in nature which leads to ambiguities in the classification decision boundaries of

several digits.

As shown in 3 we make use of KL divergence to improve the outputs. Since KL leads to entropy minimization between teacher and student we propose to use it. In Appendix Section D specifically D.2 we illustrate the process of improving it through usage of KL between the teacher and student outputs. We observe better results for several IMs for ImageNet-1k dataset and Double-colored MNIST dataset. In addition, we also present techniques that didn't work as expected such as usage of L2 loss, cross entropy for the GAN network, usage of KL and L1 loss for every activation layer of generator etc.

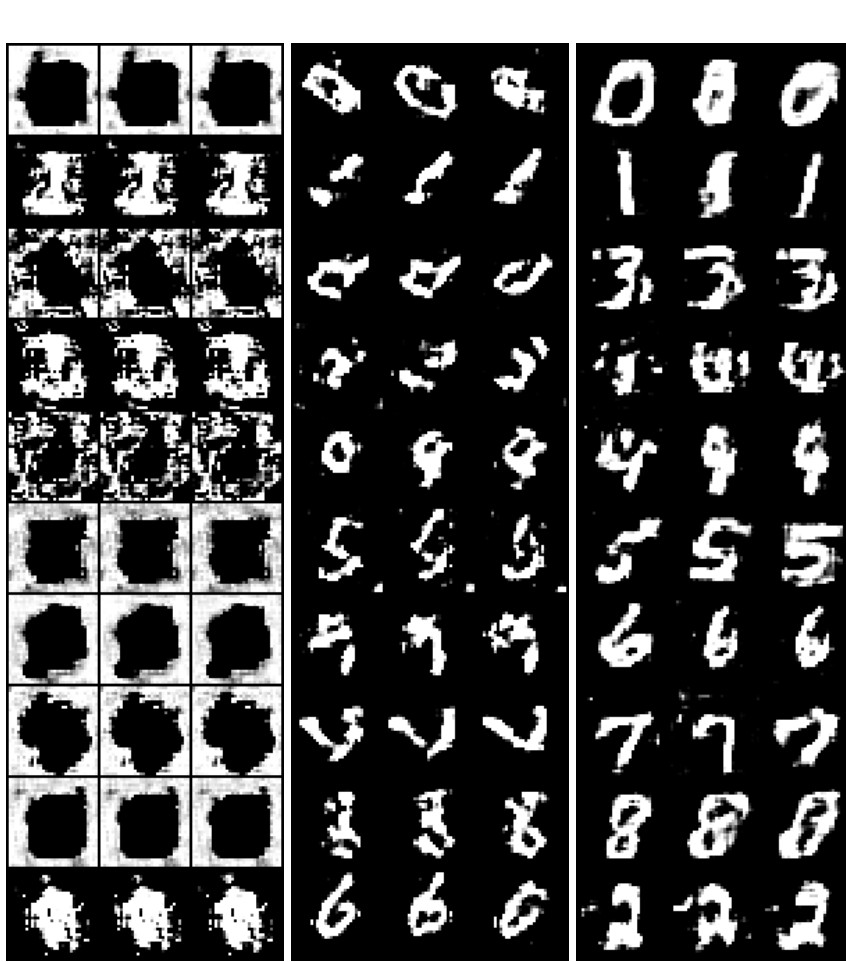

Fig. 3: Images generated with modified CGNs where, respectively, unit Gaussian noise, random rotation in a 180 degree range, and 0.75 mask transparency were introduced

## 4.5    Additional results: study of the shape IM

To evaluate the effects of the IMs, we trained several modified CGN models and used images generated by them to train classifiers. We primarily focused on the shape IM. We added Gaussian noise, introduced transparency and randomly rotated the shape mask, after which data was generated to train a classifier that distinguishes real images from counterfactual ones. This analysis was done using the Double-colored MNIST dataset. Sample shape masks generated from this process are displayed in Figure 3, and the test accuracies in Table 1. It is apparent that the test accuracy on the noise and rotation adjustments are very low, which is most likely the result of overfitting on the training set, as the train accuracies for these adjustments are quite high. The test accuracy for transparency is considerably higher. The shape masks of this adjustment are more akin to the masks achieved using regular CGNs. The other mask shapes are very different, and they could potentially be used to make classifiers more robust when mixed with regular data during training. Because this is an extensive topic, we believe it warrants further research.

|                | Noise | Rotation | Transparency |
|----------------|-------|----------|--------------|
| Train Accuracy | 99.9  | 99.1     | 94.7         |
| Test Accuracy  | 14.96 | 13.51    | 58.86        |

Table 1: Classifier test results for shape IM analysis. The classifier predicts which images are generated by CGNs and which are real.

## 5    Discussion and conclusion

Through our experiments we show that the process of Knowledge Distillation is not limited to the task of transferring classification or image generation knowledge to a less parameterized (low model capacity) network, but can also be used for the task of sampling from different manifolds that are possibly synthetic manifolds but not true manifolds such as shape, texture and background through a simple and effective approach. Interestingly, we only need access to the pretrained source model to transfer the knowledge while ignoring factors like details about the training process and background knowledge about counterfactuals and causality, etc. With the prevalence of heavily parameterized models such as BigGAN and DALL·E 2, it is often hard to even load the model in inference stage on some devices. By introducing new terms such as KL divergence in the existing knowledge distillation we are able to generate better results for some IMs for both Imagenet-1k and Double-colored MNIST datasets. With the usage of Knowledge distillation one can transfer the same capacity/ability to a low capacity network and still be able to obtain similar results while running it on a standard low configuration single GPUs.

# 6   Future work

Given ample time we wish to work on the following sections: improve the image generation process using high order activation functions since our data consists of rich image data, improving the teacher-student architecture process by introducing additional loss functions, usage of a neural network based composition function instead of it being analytical in its design.

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
