# OpenReview forum: "SKDCGN: Source-free Knowledge Distillation of Counterfactual Generative Networks using cGANs"
_thecvf.com/ECCV/2022/Workshop/VIPriors — VIPriors 2022 OralPosterTBD_

### Official Review · Reviewer_v3Nb · 2022-07-20
**This paper studies an interesting task: learning less parameterized models from the bigGAN teacher using knowledge distillation. However, the paper itseld lacks clarity, especially in the exps.**

**Rating:** 4
**Confidence:** 2

**Review:**


This paper aims to learn lightweight models from the bigGAN teacher using knowledge distillation (KL loss at both pixel/feature levels). The Introduction, Related work, and Approach are easy to follow and well explained.

Unfortunately, I find the exp part poorly written, making it extremely hard to understand: does the result support the claim? I lost track completely there. The same also happens in the appendix. Please add concrete conclusions or take-aways for each figure and re-write the exp part.

Despite the fact the task/problem addressed in the paper is both theoretically and practically meaningful, I would still recommend 'reject' given the flaws in the writing, especially the exp part.



Fig 1: left: are the modules in blue non-trainable? right: are the tinyGANs trainable during inference? Or there is a mistake in legend?
line 223: Abuse of notation: "S" refers to the student model in eq2, while in eq4 it is defined as the generator

line 344: what is the take-away from fig 2. I do not see a conclusion? It seems the SKDCGN generates less impressive results than the CGN.


line 351: "We realize that the student is as good as the teacher." where does this conclusion come from?

line 356/358: Grammarly incorrect. I do not follow these claims. It seems this part (exp 4.4) is not in a good shape as a submission.

Appendix B2/B3: It is unclear what the takeaways are in the figures? I find it hard to interpret

appendix: line 226/227 typos.

---

### Official Review · Reviewer_8nMo · 2022-07-27
**A knowledge distillation method with limited novelty**

**Rating:** 6
**Confidence:** 3

**Review:**

[Summary]
This paper presents a method that trained three tinyGANs from the Counterfactual GAN (CGNs) for shape, background and texture independent components respectively. The main novelty of this paper is the combination of the TinyGANs [7] and CGNs [3]. The author shows it is feasible to train the proposed method in the knowledge distillation way.

[Paper strength]
- The paper contributes to the combination of the TinyGANs [7] and CGNs [3]. The TinyGANs[7] train the one TinyGANs from one BigGANs, and the CGNs trained three BigGANs [4] for object shape, texture and background respectively. The proposed method in this paper trained three TinyGANs (using the way in [7]) from three BigGANs in the CGNs [3].

-  The secondary contribution is the author uses the KL divergence as the additional loss compared to [7]. In section 4.4, the author shows that using KL divergency could improve performance.

- In the experiment, the author shows that the proposed method could generate reasonable results on ImageNet and MNIST datasets.

[Paper weakness]
- The novelty contribution is limited. The proposed method is to train the TinyGANs [3] from the CGANs [3] model. The proposed architecture shown in Fig. 1 is similar to [7] except it is for three separated models, and the three separated models are introduced by [3]. The loss terms 1 to 3 are from [7] with the exact same equation and adapted text, and the loss term 4 is from [21]. I did not find any interesting technical contribution.

- The experiment is limited in terms of the error metric and conclusion. There are only qualitative results for the main experiments in sections 4.3 and 4.4. The comparison between the proposed method and baseline can only be judged subjectively. In the related works [7], there are other metrics such as FID to compare different methods. From these qualitative results, I can only judge that the proposed method can generate the somehow similar results to the baseline but for sure is worse than the baseline. Compared to the results achieved in [7], the TinyGAN in [7] has comparable results with the BigGAN model.

- Inappropriate baseline. The baseline method is CGN but each BigGAN is replaced by a TinyGAN. However, the author motivates the proposed method that the BigGANs in the original CGN is over-parameterrized (line 45). I expected the author would compare their method to the original CGN,  too. If we can train the CGN with the TinyGAN, I think we do not have the over-parameterized networks problem.

Other minor issues:
- I do not find the necessarily of section 4.5.
- Misuse of the supplementary material. The author refers to supplementary on improving the method in section D. However, according to the author guideline of ECCV (https://eccv2022.ecva.net/submission/call-for-papers/), "Reviewers will be encouraged to look at it (supplementary), but are not obligated to do so", and "It may not include results obtained with an improved version of the method".

---

### Official Review · Reviewer_hcve · 2022-08-02
**Preliminary work with interesting future extensions**

**Rating:** 6
**Confidence:** 4

**Review:**

The paper describes an approach for developing resource-efficient Counterfactual Generative Networks (CGNs) through Knowledge Distillation (KD) from black-box pre-trained CGNs to smaller TinyGANS. The approach is then named by the authors Source-free
Knowledge Distillation of Counterfactual Generative Networks (SKDCGN).
The authors employ a TinyGAN for each of the independent mechanisms (i.e., shape, texture, background) in order to increase modularity and reduce the size of the overall model.

Pros:

- The paper is eligible for the workshop since the concept of “prior” is applicable both to 1) the knowledge transfer from the teacher CGN and 2) the built-in approach of CGNs that employ inductive biases (i.e., shape, texture, and background) to generate realistic images.

- The paper tackles two critical problems of modern deep learning literature, i.e., 1) reducing the size of state-of-the-art GANs and 2) learning from large pre-trained models through black-box access.

- The paper is well written and clearly presents the objectives, methodology, and qualitative results.

- The related work section provides a good overview of the literature concerning this paper (i.e., CGNs and KD).


Cons:

- The paper employs and combines published techniques [3, 7] rather than proposing a novel method.

- The evaluation of the approach is preliminary and needs extensions.

  Section 4 mainly presents qualitative rather than quantitative results. It would be interesting to evaluate SKDCGN on Out-of-Domain (OOD) classification tasks as performed in [3]. The results would probably be less
  compelling but still interesting to observe.

  Furthermore, given that the objective of SKDCGN is to make current CGNs more lightweight, it would be better to report a plot/table in
  which metrics quantify improvements. For instance, the reduction of the number of trainable parameters, GPU memory usage, etc.

- As visible in Fig. 2, the texture mechanism is the one that suffers more from the reduction of size. Textures of Fig. 2 (b) generated by
  SKDCGN are hardly distinguishable.
  On the contrary, the shape and background mechanisms mimic quite well the original generations.
  Future work should develop more on the texture generation mechanism. For instance, by including data augmentation or other
  approaches that improve image synthesis for GANs.


Minor issues:

- The abstract is a bit lengthy and could probably be pruned.

- Is the legend of Fig. 1 partially wrong? During inference, the TinyGANs should be “fixed” and during training “trainable”. Further, the composition mechanism is said to be untrainable by default.

---

### Decision · Program_Chairs · 2022-08-08

**Decision:**

Accept (Oral/Poster TBD)

**Comment:**

Dear authors,


Congratulations! Your work has been accepted to the VIPriors workshop. Decisions on oral/poster presentations will follow later, when the program of the workshop is finalized.

*Please note the first action item is due on Wednesday! Please see instructions below.*

**Camera-ready instructions**

There is some work left to be done to ensure your work is included in the ECCV conference workshop proceedings. The ECCV publication managers use CMT to collect all workshop papers. This means we will migrate your paper from the VIPriors OpenReview page to the centralized ECCV workshop proceedings CMT page. The VIPriors program committee will ensure the details of your work (name, title, email address) are transferred to the CMT page, after which the ECCV proceeding managers will invite you to upload the camera-ready version of your work to the centralized ECCV CMT workshop proceedings page.

Please carefully follow the following instructions:
- **Before August 10th**, ensure that the first author has a CMT account under the same email address as the OpenReview account through which the accepted work was submitted. This account will be used to invite you to upload the camera-ready paper.
- Fill out this form, to inform us that the CMT account is in order: https://docs.google.com/forms/d/e/1FAIpQLSfyAoPv2_srESKaLRHIsHoWe3Fss1Z50ykdH7SzZpenA0m_5g/viewform
- Await instructions from the ECCV publication organizers, sent through CMT, on how to submit your camera-ready paper.
- Submit the camera-ready paper **before August 22nd**. Follow the camera-ready instructions for the main conference: https://eccv2022.ecva.net/submission/call-for-papers/.

**Attending the workshop**

We invite all authors of accepted works to attend the workshop in person on October 24th 2022 at ECCV in Tel Aviv. Please note a conference registration is required to attend the workshop. The workshop will be hybrid, enabling both in-person and remote attendance. We hope all accepted works can be represented in-person by at least one author, but we understand if this is not possible. Remote attendance of the workshop will be possible, though unfortunately there are limits on presenting works remotely: we intend to enable remote oral presentations, but this is not possible for posters.

Please fill out this form *before September 26th* to inform us of your attendance: https://docs.google.com/forms/d/e/1FAIpQLSfqRhdd2pq8t4CC8hL_c8fQo_TWcbzuQH3KGLzKVE36iTW_oQ/viewform.

**Presenting your work at the workshop**

Authors of all accepted papers are invited to present a poster at the workshop. Instructions on poster format will follow at a later date, but we will ask you to print and bring your own poster to the workshop.


For more information, as well as updates on the program of the workshop, keep an eye on our website: https://vipriors.github.io.

We thank you for choosing to submit to our workshop, and we are very much looking forward to hosting you in person in Tel Aviv!


Kind regards,

Robert-Jan Bruintjes
VIPriors program committee